# Alterations in Carbohydrate Quantities in Freeze-Dried, Relative to Fresh or Frozen Maize Leaf Disks

**DOI:** 10.3390/biom13010148

**Published:** 2023-01-11

**Authors:** Lynnette M. A. Dirk, Tianyong Zhao, John May, Tao Li, Qinghui Han, Yumin Zhang, Mohammad R. Sahib, Allan Bruce Downie

**Affiliations:** 1Department of Horticulture, Seed Biology Group, College of Agriculture, Food and Environment, University of Kentucky, 1405 Veterans Drive, Lexington, KY 40546, USA; 2State Key Laboratory of Crop Stress Biology for Arid Areas, Department of Biochemistry and Molecular Biology, College of Life Science, Northwest Agriculture and Forestry University, Yangling, Xianyang 712100, China; 3Department of Plant and Soil Sciences, College of Agriculture, Food and Environment, University of Kentucky, N-222A Ag Science North, Lexington, KY 40546, USA; 4Collaborative Innovation Center of Henan Grain Crops, State Key Laboratory of Wheat and Maize Crop Science, College of Life Science, Henan Agriculture University, Zhengzhou 450002, China; 5The Key Laboratory for Quality Improvement of Agricultural Products of Zhejiang Province, College of Agriculture and Food Science, Zhejiang Agriculture and Forestry University, Hangzhou 311300, China; 6College of Agriculture, Al-Qasim Green University, Babylon 00964, Iraq

**Keywords:** carbohydrate, galactose, glycerol–galactopyranoside, lyophilization, raffinose family oligosaccharides

## Abstract

For various reasons, leaves are occasionally lyophilized prior to storage at −80 °C and preparing extracts. Soluble carbohydrate identity and quantity from maize leaf disks were ascertained in two separate years using anion exchange HPLC with pulsed electrochemical detection. Analyses were made from disks after freezing in liquid nitrogen with or without subsequent lyophilization (both years) or directly after removal from plants with or without lyophilization (only in the second year). By adding the lyophilizing step, galactose content consistently increased and, frequently, so did galactoglycerols. The source of the galactose increase with the added lyophilizing step was not due to metabolizing raffinose, as the *raffinose synthase* (*rafs*) null mutant leaves, which do not make that trisaccharide, also had a similar increase in galactose content with lyophilization. Apparently, the ester linkages attaching free fatty acids to galactoglycerolipids of the chloroplast are particularly sensitive to cleavage during lyophilization, resulting in increases in galactoglycerols. Regardless of the galactose source, a systematic error is introduced for carbohydrate (and, most likely, also chloroplast mono- or digalactosyldiacylglycerol) amounts when maize leaf samples are lyophilized prior to extraction. The recognition of lyophilization as a source of galactose increase provides a cautionary note for investigators of soluble carbohydrates.

## 1. Introduction

Non-reducing sugars, such as trehalose, sucrose, and the raffinose family oligosaccharides (RFOs), accumulate, along with intrinsically disordered proteins (IDPs), prior to seed maturation and desiccation. Such accumulation has been positively and tightly correlated with a successful entry into anhydrobiosis [1,2,3,4]. In plants, these non-reducing sugars and IDPs, namely the Late Embryogenesis Abundant Proteins (LEAPs), are also suggested to protect vegetative plant organs from dehydration stress. They can replace water, serve to maintain the complex molecular structure of lipids and proteins [5,6], and also reduce reactive oxygen species (ROS) concentrations [7,8]. In plants, the LEAPs are thought to act synergistically with specific ratios of non-reducing oligosaccharides to enhance drought and desiccation tolerance in mechanisms that are incompletely understood [9,10,11,12].

Advancing our understanding of carbohydrate status in maize leaves to create resilience to suboptimal water status requires, first and foremost, analyses that represent, as close as possible, actual leaf cell contents at the time of sampling. Literature reports of maize sugar contents are based on either fresh- (e.g., [13,14]) or dry-weight (e.g., [15,16]). The removal of water via lyophilization after freezing the leaf sample in liquid nitrogen is commonly how a dry weight is obtained prior to the extraction of soluble carbohydrates.

In our work extracting carbohydrates from liquid nitrogen frozen or liquid nitrogen frozen and then lyophilized leaf material, we have noted that the recovery of certain carbohydrates differed. Such changes were reported previously for freeze- versus microwave-dried samples [17]. Such alterations have led investigators to suggest that some carbohydrates may be tightly bound to leaf components [18] or that some oligosaccharides are metabolized due to problems maintaining samples completely frozen during lyophilization [19]. We needed to further examine the correlation between how the tissue was treated (after sampling but prior to preparing the extract) and the sugar contents. Using wild type (W22), null segregant (NS), and *raffinose synthase* (*rafs*) mutant maize (*Zea mays*) leaf tissue, we have investigated carbohydrate amounts between fresh-, frozen-, and lyophilized-leaf disks for any discrepancies due to the disk treatment. In this comparison, the carbohydrate identity and amounts in leaves frozen in liquid nitrogen were taken as most representative of the true cellular carbohydrate profiles. These were compared to treatments that included lyophilization with or without prior freezing in liquid nitrogen.

## 2. Materials and Methods

Plant material: Maize seeds (wild type (WT W22), null segregant (NS W22), or *raffinose synthase* (*rafs*) [EC 2.4.1.82; GRMZM2G150906; UFMu-09411] mutant W22) previously acquired from the Maize Genetics Stock Center, Urbana, IL, USA [16,20], were placed on two layers of moist germination blotter (Anchor Paper Company, St. Paul, MN, USA) in germination boxes which were kept at 25 °C in the dark. Following the completion of germination, when coleoptiles were one inch or longer, seedlings were carefully transplanted to 5-gallon pots filled with three parts Promix (Premier Tech., Rivière-du-Loup, Québec, Canada) and one part sterilized, clay field soil. Seedlings were watered into the soil to eliminate air gaps around seedling roots and seeds, and water emitters were placed in each pot. Automatic watering was performed for 10 min every 12 h. Greenhouse settings were 30 °C with 16 h of supplemental light (Corn cob LEDs, ReneSola, 100 W) and 25 °C with 8 h of darkness. Once the third leaf had emerged, 2 mL of Marathon 1% G (granular; Olympic Horticultural Products, Bluffton, SC, USA) was sprinkled on the top of the soil. Fertigation occurred every third day with Peter’s Professional 20:20:20 fertilizer (ICL Specialty Fertilizers–Americas, Summerville, SC, USA). Upon silking, the 9th and 10th leaves formed (from the base of the plant) were sampled from six different plants of each genotype (null segregant (NS) or *raffinose synthase* (*rafs*) in year 1 expanded to include WT W22 in year 2; Figure 1).

Harvest, freezing, and freeze-drying: Upon tasseling, pairs of 3.7 cm (dia.; 11.62 cm^2^) disks were cut from the blades of each leaf of a plant on either side of the main vein 20 cm down from the tip (duplicates; Figure 1). These disks were taken from the 9th and 10th leaf from the same plant (replications) and one disk from the 9th and its replicate disk from the 10th leaf underwent the same treatment. The 9th and 10th leaves are vigorous at tasseling and large enough to allow the acquisition of two circular samples beside each other on either side of the main vein. The main vein is avoided to minimize assimilate. In the first year, the constituted treatments were frozen in liquid nitrogen (LN_2_; Fz) or frozen in LN_2_ and then freeze-dried (FzL), and disks of both treatments were kept at –80 °C until processing. In the second year, disks were harvested and immediately ground, and carbohydrates were extracted (processed fresh; Fr) or harvested and immediately freeze-dried (FrL) and then kept at −80 °C until processing. Other disks were frozen in LN_2_ and then kept at −80 °C until processing (Fz) or frozen in LN_2_, freeze-dried (FzL), and then kept at −80 °C until processing (Figure 1). Disks were lyophilized by placing them in a 50 mL Falcon tube with the cap removed, using a loosely balled Kimwipe to hold the disk in place; then the tubes were placed in a lyophilizer (Bench Top Model 75035, LabConCo, Kansas City, MO, USA) with a condenser (cold finger, −108 °C), and a vacuum drawn (between 100 and 5 micrometers Hg). Leaf disks were lyophilized until no more ice formed on the cold finger. Once leaf disks were desiccated, the Falcon tubes were retrieved from the lyophilizer, quickly capped, and placed at −80 °C until processing.

Sample processing: Other than the fresh samples in year 2, leaf disks in their separate 50 mL Falcon tubes were retrieved from −80 °C and immediately placed in LN_2_. Tubes were retrieved from LN_2_ one at a time, opened, and the disk dropped into a mortar filled with LN_2_. Disks were ground using a pestle with more LN_2_ carefully added as it boiled off until the entire disk was pulverized. Ethanol (1 mL 70% EtOH), that is 1 mM with respect to 2-deoxyglucose (2-DOG; internal standard; Sigma-Aldrich, St. Louis, MO, USA), was added to the powder, and a slurry was created as the liquid thawed. A wide-bore pipet was used to transfer this slurry to a 15 mL polypropylene centrifuge tube (Corning Inc., Corning, NY, USA). An additional 1 mL of 70% EtOH without 2-DOG was added to the mortar, and the remaining slurry was ground and transferred to the centrifuge tube. Washing the mortar and pestle in this manner was repeated with 70% EtOH three more times for a total of 5 mL of 70% EtOH. The tubes and contents were centrifuged (13,000× *g*, 10 min, 4 °C), and the supernatant was transferred to 50 mL Falcon tubes. Water (20 mL) was added to bring the contents to 25 mL, reducing the ethanol content such that the sample would remain frozen in the lyophilizer (to prevent bumping). This tube was placed at −80 °C on a 45 angle (increasing the frozen sample surface area) to rapidly freeze, rather than plunging the tubes in LN_2_, which had previously occasionally cracked the tube, spilling the contents. Once frozen, the caps were quickly removed from the tubes, and they were placed in the lyophilizer. When required, the vacuum was released, ice was removed from the cold finger and trap, the samples were reintroduced, and the vacuum was reapplied without thawing the sample. This continued until the contents of each tube were dry. The powder was reconstituted with 1 mL of water and kept on ice for 12 h with intermittent vortexing. The liquid was transferred to a microcentrifuge tube, centrifuged (13,000× *g*, 10 min), and the supernatant was filtered through a 3.0 μm glass fiber prefilter, 0.2 μm Supor filter sample clearance plate (AcroPrep™ Advance Filter Plates for Lysate Clearance, Pall Corporation, Port Washington, NY, USA). The eluate (100 μL) was diluted in 900 μL of water in a vial and introduced into an autosampler.

Carbohydrate analysis: A Dionex HPLC (ICS-6000DC, SP; Dionex/ThermoFisher, Waltham, MA, USA) with pulsed electrochemical detection (PED) was used to run two isocratic elution protocols, one for mono- and di-saccharides (18 mM NaOH) and the second for melibiose, raffinose, and maltose (156 mM NaOH, 2.5 mM sodium acetate; Appendix A). Dionex PA1 guard and analytical columns were used in tandem to resolve carbohydrates using factory-recommended PED settings for carbohydrates and amino acids. An autosampler (10 °C sample tray; Dionex AS-AP) introduced carbohydrate standards or samples onto the column using a 25 μL sample loop. Each run was coordinated using Chromeleon Software, v. 7.2.9 (Dionex/ThermoFisher). Calibration of elution times and confirmation of the linearity of detector response over a 10-fold carbohydrate concentration was performed using three different standard concentrations while forcing the calibration line through zero (Chromeleon). Upon reintegration of elution times, baselines, and peak identities (Chromeleon; Appendix A), the spreadsheets were exported to Excel. Carbohydrate quantities were adjusted for 2-DOG recovery (internal standard) for sample loop size and dilution.

Unknown identification: There were two unknown peaks that tended to vary depending on leaf disk lyophilization. Three samples, varying in the amount of these unknowns in traces from the Dionex HPLC (Appendix A), were diluted in 80% acetonitrile, 20% water, and 0.1% formic acid, and they were analyzed by hydrophilic interaction chromatography (HILIC) on a Waters Acquity UPLC (Waters Corporation, Milford, MA, USA) coupled to a Synapt G2 Q-ToF mass spectrometer (Waters Corp.). Chromatographic separation was obtained using a Waters BEH Amide UPLC column (1.7 mm, 2.1 mm × 150 mm) at 30 °C and a custom elution protocol (Appendix A) with elution times provided in Appendix A. The high-resolution mass spectrometer was operated in negative ion electrospray mode and scanned from 50 to 1000 Da in 0.3 s. Leucine enkephalin was used to provide a lock mass (*m*/*z* 554.2615).

By comparing the areas of peaks across the three different MS chromatograms, two peaks of interest were identified (Appendix A) as possible unknowns identified in the Dionex system (Appendix A). The first peak was at a retention time of 3.70 min with a [M–H]^−^ ion mass of 253.092, and the second peak was at a retention time of 5.86 min with a [M–H]^−^ ion mass of 415.146 (Appendix A, respectively). Elemental composition searches from the accurate masses were performed with the most probable formula fits producing C_9_H_18_O_8_ and C_15_H_28_O_13_, respectively, for the neutral masses. The first unknown was tentatively identified as glycerol-O-β-D-galactopyranoside (GG) and purchased (Cayman Chemical Company, Ann Arbor, MI, USA). The commercial compound was subjected to liquid chromatographic analysis using both the Dionex and Waters systems with retention times matching that of the first unknown. The [M − H]^−^ mass of the compound was also verified using the Synapt G2 Q-ToF mass spectrometer attached to the latter system.

Enzymatic hydrolysis of maize leaf extracts: An aliquot of a sample from the *raffinose synthase* mutant (removing raffinose as a confounding source of galactose upon hydrolysis with α-galactosidase) containing large amounts of unknowns 1 and 2 was spiked with GG and subjected to HPLC analysis (Figure 2A). Aliquots of the same sample were digested with either α-galactosidase (EC 3.2.1.22, from green coffee beans, Sigma (G-8507), St. Louis, MO, USA) or β-galactosidase (EC 3.2.1.23, from *Escherichia coli*, Sigma G-5635), or both, at 37 °C for 12 h before being diluted and analyzed. The activity and specificity of the commercial enzyme preparations were tested using positive controls containing aliquots of raffinose (O-α-D-galactopryranosyl-(1→6)-α-D-glucopyranosyl-β-D-fructofuranoside; United States Biological Inc. Life Sciences, Salem, MA, USA) or lactulose (4-O-β-D-galactopyranosyl-D-fructofuranoside; Sigma).

Correlation between water potential and carbohydrate amount: Turgid maize leaves and leaves at various stages of water loss, at the same position on the plant as those used for carbohydrate analysis, were wetted with water and slightly abraded with an emery board according to instructions provided with the WP4-T Dewpoint Potentiometer (Decagon Devices, Pullman, WA, USA). Leaves were patted dry, and leaf disks were taken from these abraded regions, placed in round plastic sample cups, and assessed for water potential. Once assessed, covers were added to the cups, the water potential was written on them, and the cups floated on LN_2_. The cups were transferred to −80 °C until they were retrieved for processing for carbohydrate analysis as detailed previously.

Statistical analysis: The carbohydrate amounts from the pairs of leaf disks from leaves 9 and 10, treated in the same manner (Figure 1A), were averaged among plants. Carbohydrate amounts for each of the combined leaf disk pairs from the plant replicates were subjected to analysis of variance (ANOVA; Statistical Analysis System (SAS); Cary, NC, USA), testing the non-directional hypothesis that there was a statistically significant difference in at least one of the carbohydrate amounts between non-lyophilized and lyophilized tissue. The null hypothesis was that there was no difference in any carbohydrate amount between leaf disks that were or were not subjected to lyophilization. If significant differences were indicated, the analysis was re-run and the averages were compared using Scheffe’s multiple pairwise comparison at an experiment-wide error rate of α = 0.05. Figures were generated from CAS SciFinder^n^ (CAS, Columbus, OH, USA), ChemDraw Professional (PerkinElmer Informatics, Inc., Hebron, KY, USA), or SigmaPlot for Windows Version 14.5 (Systat Software Inc., Point Richmond, CA, USA).

Water potential and carbohydrate amount data from leaf disks taken from turgid or wilted leaves were plotted using Proc plot (SAS) to assess patterns in carbohydrate amounts as leaf water potential decreased. When trends in the data were ascertained, a simple linear regression using Proc reg (SAS) was used to determine the significance of the model, predict the adjusted R^2^, and reveal the nature of the relationship (positive or negative).

A stepwise progression was used to ascertain if there were relationships between water potential or any carbohydrate and galactose, GG, or unknown 2 amounts (Proc Reg, SAS) to determine which of the gathered parameters was influencing the amounts of these metabolites the most. To be considered for inclusion in the model, the predictive value had to be 0.05 or less (Proc Reg, significance level entry “sle” = 0.05), and the same criterion was imposed on whether a parameter was retained in the model (significance level stay “sls” = 0.05) as a new predictor was added.

## 3. Results

Carbohydrate recoveries varied depending on the treatment of the leaf tissue disk prior to sample processing. Of the unknowns present in maize leaf disks with the extraction and analyses performed (Appendix A) that varied depending on lyophilization, one was determined to be glycerol-O-β-D-galactopyranoside (GG). The basis of the identification is fourfold. First, the unknown 1 co-eluted with a commercial source using two different liquid chromatography systems (Figure 2A; Dionex system; Appendix A; Waters system). Second, an identical neutral, monoisotopic mass to the commercial source (254.23; Molecular formula: C_9_H_18_O_8_) was determined using a Synapt G2 QTOF (Appendix A). Third, the unknown 1 was stable to α-galactosidase activity. Finally, the unknown 1 was hydrolyzed by β-galactosidase as its peak from the chromatogram was missing and simultaneously the peak height of galactose increased (Figure 2 B,C and Figure 3A–C).

The unknown 2 metabolite (neutral, monoisotopic mass of 416.153; Appendix A; one possible molecular formula: C_15_H_28_O_13_) had at least one α-linked galactose based on hydrolysis by α-galactosidase (Figure 2B,C and Figure 3A–C). This galactose is apparently attached to a β-linked galactose because treatment with β-galactosidase does not hydrolyze unknown 2 unless it has been previously hydrolyzed by α-galactosidase (Figure 3A–C). Hydrolysis of unknown 2 with α-galactosidase removes the unknown 2 peak but increases the peak height of GG (Figure 2B, C and Figure 3A–C). Furthermore, treatment with both α- and β-galactosidase increases galactose amounts beyond that achieved by α-galactosidase hydrolysis of unknown 2 and galactinol or β-galactosidase hydrolysis of GG (Figure 3A–C). Based on these observations, it is possible that unknown 2 might be 2,3-dihydroxypropyl 6-*O*-α-D-galactopyranosyl-β-D-galactopyranoside (drawn in Figure 3B) from lipase action on 1,2-diacyl-3-O-(α-D-galactosyl(1→6)-O-β-D-galactosyl)-sn-glycerol. The stability of unknown 2 when treated with β-galactosidase can be explained by the catalytic mechanism of this exo-enzyme [21], that is, the beta-linked galactose is shielded from the enzyme by the alpha-linked galactose. Simultaneous digestion of a sample with large amounts of unknown 2 with both α- and β-galactosidase eliminated unknown 2, GG, and galactinol while concurrently increasing galactose amounts greater than either α- or β-galactosidase working alone on the same extract (Figure 3A–C). We could not acquire digalactosyl–glycerol commercially and so this identification is tentative. Standards were amended to include GG (Figure 2 and Figure 3; Appendix A).

Regardless of year, genotype, or fresh or frozen tissue, lyophilized leaf disks consistently had statistically significantly greater amounts of galactose than non-lyophilized leaves (Figure 4, Figure 5, Figure 6, Figure 7, Figure 8, Figure 9, Figure 10 and Figure 11A). Apart from the first year in *rafs* tissue, both GG (Figure 4, Figure 5, Figure 6, Figure 7, Figure 8, Figure 9, Figure 10 and Figure 11B) and unknown 2 (Figure 4, Figure 5, Figure 6, Figure 7, Figure 8, Figure 9, Figure 10 and Figure 11D) increased significantly after lyophilizing. In the second year, sorbitol was statistically significantly greater in lyophilized leaves, regardless of genotype or whether disks had been exposed to liquid nitrogen (LN_2_) or not prior to lyophilizing (Figure 6, Figure 7, Figure 8, Figure 9, Figure 10 and Figure 11C). In the second year, galactinol was usually equal (Figure 7B and Figure 9B) or statistically significantly greater (Figure 6B, Figure 8B and Figure 11B) in lyophilized relative to non-lyophilized tissues, except for *rafs* leaf disks without prior LN_2_ exposure, where it was statistically significantly less when lyophilized (Fr versus FrL; Figure 10B). Generally, raffinose, when present, tended to decrease after lyophilization, sucrose remained the same, and both glucose and fructose tended to increase, occasionally significantly (glucose in Figure 9C).

Both GG and unknown 2 usually increased in abundance after lyophilization. To determine if this increase could be mimicked as leaf cells decreased in water potential, we sampled detached leaves that were at various water potentials and assessed quantities of GG and unknown 2. While both metabolites were significantly associated with water potential, their adjusted R^2^ values for water potential were lower than the adjusted R^2^ values for water potential of galactose, glucose, and fructose (i.e., these carbohydrates correlated more strongly with leaf water potential). Furthermore, GG and unknown 2 were highly correlated (Appendix A). We, therefore, attempted to determine possible predictors of the accumulation of GG or unknown 2 from the data we had gathered. Stepwise linear regression, limited to the introduction and retention of variables into the model at a significance level of 0.05 or lower, respectively, determined that the best predictors of GG amounts were unknown 2, fructose, and mannitol, while GG, fructose, mannitol, and trehalose were the best predictors of unknown 2 amounts (Appendix A).

## 4. Discussion

Two unknown maize leaf metabolites were determined to frequently increase after lyophilizing samples, relative to non-lyophilized tissue, regardless of genotype or prior exposure to LN_2_. One of these was identified as glycerol-O-β-D-galactopyranoside (GG), a lipase degradation product of the chloroplast membrane lipid, monogalactosyl diacylglycerol (MGDG) [22] that, along with digalactosyl diacylglycerol (DGDG), are the two most abundant membrane lipids in the maize chloroplast [23]. There are many different lipid–acyl hydrolases that were identified that act to cleave esterified free fatty acids from MGDG [24], one of which (EC 3.1.1.26) also works on DGDG [25,26]. Lipase-mediated removal of acyl chains from GG during lyophilization may be a source of GG (and glycerol digalactopyranoside if that is truly the identity of unknown 2). Certainly, the galacto–lipase activity, responsible for free fatty acid production from DGDG and MGDG, and with that, off flavors, in the vegetable processing industry, is a target of investigation as a marker enzyme for assessing vegetable blanching efficacy [27,28]. It is our assumption that the source of the two galactoglycerols is from the degradation of thylakoid membrane components; as the only other biochemical pathway for their synthesis currently known would be the energy-intensive galactose addition from UDP-galactose to diacylglycerol [29]. While it is possible that these galactose-containing lipids are also the source of galactose increase after lyophilization resulting from β-galactosidase (acting on monogalactosyldiacylglycerol) or α- then β-galactosidase (acting sequentially on digalactosyldiacylglycerol), a definitive determination of the source(s) of galactose after lyophilizing remains elusive. Whatever the source of GG, it is quite possible that unknown 2 may be from the same source (i.e., chloroplast membrane components after lipase action) as the abundance of both GG and the unknown 2 are tightly linked (Appendix A). Digestion of the second unknown with α-galactosidase increases GG and galactose amounts, some of the latter coming from galactinol (Figure 2B,C and Figure 3A–C). However, when unknown 2 is digested with both α- and β-galactosidase, it is eliminated, while galactose abundance increases significantly more than when this extract is treated with either α- or β-galactosidase alone (Figure 3A–C). Whereas in this report, GG accumulation was associated with leaf stress due to lyophilizing, this metabolite was associated with superior performance under heat stress in wheat [30].

Discrepancies in the amount of carbohydrates present in lyophilized leaves were reported previously, although in the context of frozen tissues thawing during the lyophilization of large volumes of tissue [19]. We point out that in our experiments herein, the leaf disks presented very little mass, and that the equipment we use can maintain multiple 25 mL samples of 14% *v/v* ethanol solution frozen (when removing the diluted extraction solution prior to HPLC analysis). Additionally, cleaning the cold trap prior to re-establishing a vacuum was rapid, never resulting in the samples thawing during the leaf disk lyophilization or extraction solution removal.

We are studying the presumptive influence non-reducing sugars sucrose, raffinose, and galactinol have on protecting the longevity of desiccated seeds [16,31,32] and in reducing the unwanted effects of abiotic stress on vegetative tissues [20,33,34]. Particularly regarding dehydration of vegetative tissues, when comparing our results to those of others, it was important to determine if any eventual discrepancies might be attributable to sample treatment prior to extraction.

Apart from lyophilized leaf disks being tougher and generally more difficult to pulverize than their frozen counterparts (ABD and MRS personal observations), no general gross morphology changes were observed that might influence carbohydrate recovery from leaf tissues treated in various manners. Following preliminary assessments of leaves, an observation was made that freeze-dried samples had statistically significantly more galactose than frozen samples. This discrepancy could potentially be explained by assuming that raffinose, galactinol, or some other galactose-containing moiety was the source of galactose when some of the stores of these carbohydrates were hydrolyzed by a galactosidase during or while lyophilization. There are reports of an alkaline α-galactosidase capable of releasing the terminal galactose from the di-galactosyl glycerol polar head of the thylakoid lipid DGDG [35]. Another β-galactosidase in the chloroplast can hydrolyze galactose from monogalactosyl diacylglycerol [36,37], while a senescence-upregulated lipase capable of acting on both MGDG and DGDG has been reported [38]. There was a trend towards lower raffinose amounts in lyophilized leaf disks that were similar in amount to the increases in galactose (on a molar basis) but, while the galactose increases were statistically significant, the raffinose declines were not. Neither did sucrose increase, although there was a trend for glucose and fructose to do so. We tested the presumption that raffinose might be the source of galactose in lyophilized leaves by including in our analysis, leaves from the maize *rafs* mutant which makes no raffinose or higher-order RFOs [16]. Raffinose was eliminated in the mutant leaves, as anticipated [16] and, while galactose amounts, regardless of treatment (Fz versus FzL; Fr versus FrL) decreased overall in *rafs* leaf disks, galactose amounts were still statistically significantly greater when the disks were lyophilized.

None of the other galactose-containing carbohydrates we identified were statistically significantly reduced when the leaf disks were subjected to lyophilizing relative to the non-lyophilized disks. In fact, if there was a statistical difference, regardless of genotype, lyophilizing increased the amount of galactose-containing carbohydrate (GG, galactinol, and unknown 2). Potentially, galactose is hydrolyzed from many different sources during lyophilization. Because these sources also increase in abundance due to lyophilization (e.g., GG and unknown 2), their contributions to galactose are masked. Taken together, we are unable to determine a single source responsible for galactose increases when leaf disks are lyophilized.

## 5. Conclusions

We conclude that maize leaf samples should be collected and frozen in LN_2_ immediately without subsequent lyophilization. This is to avoid unwanted changes in soluble carbohydrate quantities, particularly both galactose and glycerol galactopyranoside (GG; potentially also glycerol digalactopyranoside) increases due to lipase action on chloroplast MGDG (potentially also DGDG) membrane components if lyophilized.

## Figures and Tables

**Figure 1 biomolecules-13-00148-f001:**
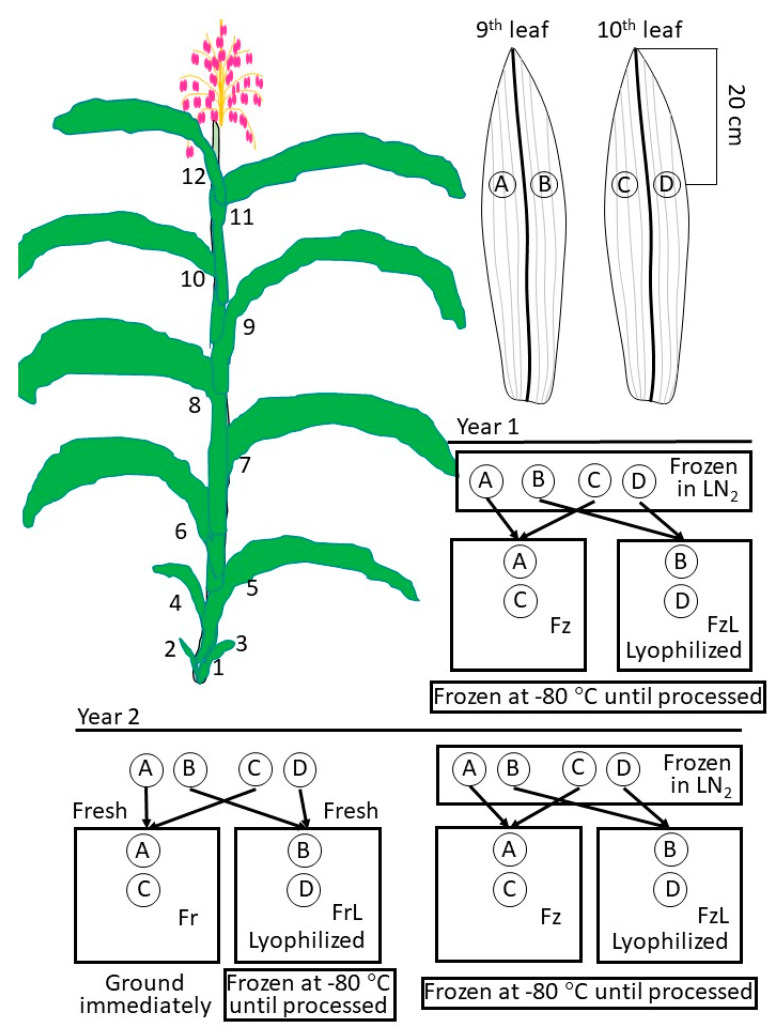
Graphic showing the sampling of the maize leaf. Maize plants (W22, *raffinose synthase* (*rafs*) [GRMZM2G150906] mutant or null segregant (NS)) were grown to maturity before leaf disks from the 9th (A and B) and 10th (C and D) leaves were sampled. One disk from the 9th and one from the 10th leaf constituted replications of each treatment. In the first year (Year 1), these leaf disks were frozen in liquid nitrogen (LN_2_) and either immediately stored at −80 °C or lyophilized to dryness and then stored at −80 °C, constituting the treatments frozen (Fz) and frozen lyophilized (FzL). In the second year (Year 2), leaf disks were harvested and immediately ground (fresh, Fr) or placed in the lyophilizer without prior freezing (FrL). Similar to the first year, other leaf disks were harvest and frozen in LN_2_ and either immediately stored at −80 °C (Fz) or lyophilized to dryness and then stored at −80 °C (FzL).

**Figure 2 biomolecules-13-00148-f002:**
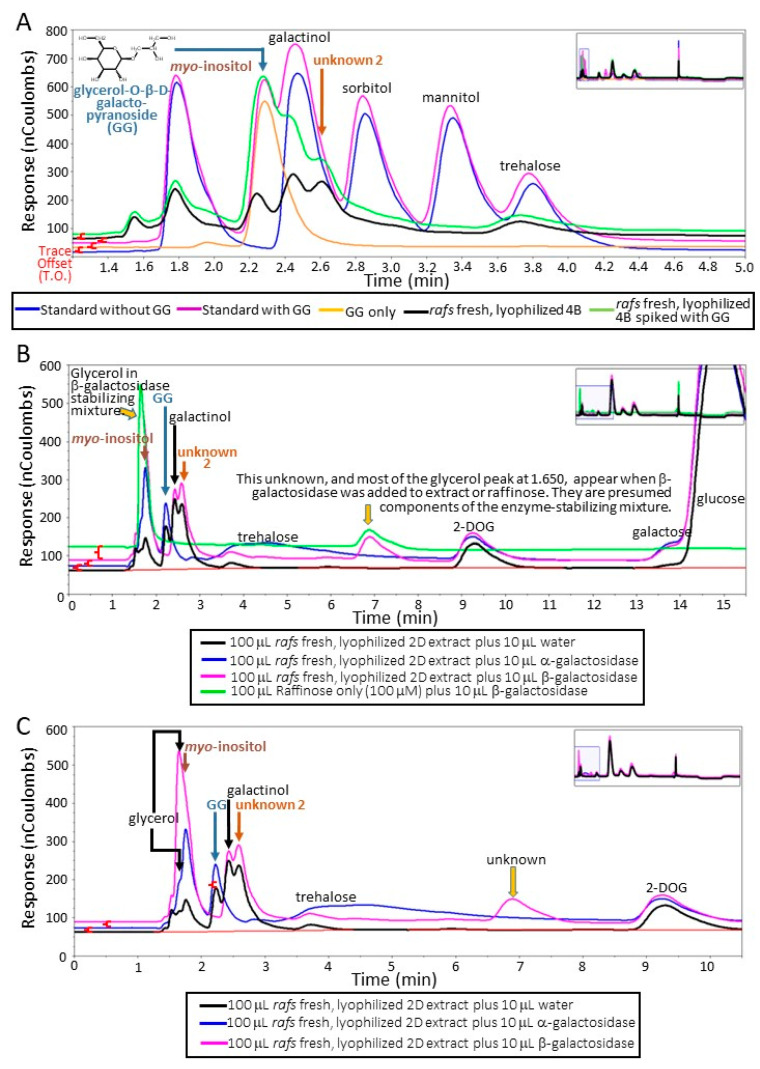
Verification of the identity of unknowns in maize leaf carbohydrate extracts. Glycerol-O-β-D-galactopyranoside (GG; inset on the left: chemical structure) was acquired from a commercial source and run on both the Waters (not shown) and (**A**) the Dionex systems. The overlaid chromatographs show the trace from a 100 µM GG sample (orange line), 100 µM GG added to the external standard (pink line) and the external standard without GG (blue line). Additionally, a sample containing large amounts of unknowns 1 and 2 (black line) was spiked with 100 µM of GG (green line) and run on the Dionex system. (**B**) Chromatographic and enzymatic verification of the tentatively identified GG as unknown 1. Overlays of the unhydrolyzed sample (black line), α- (blue line), or β-galactosidase-hydrolyzed (pink line) aliquots of a sample containing the first and second unknown. A trace of raffinose hydrolyzed with β-galactosidase is also shown (green trace) to emphasize the glycerol increase and the presence of an unknown peak eluting at ~7.0 min is predominantly due to the stabilizers/preservatives in the β-galactosidase preparation. Note the increase in galactose whenever the samples are hydrolyzed. (**C**) A closeup of the overlaid traces in B. In all figures, the trace offset (T.O.) used to overlay chromatographs (red parentheses) is insufficient to explain the peak increases in GG from spiking or enzymatic hydrolysis. This is strong evidence supporting the supposition that hydrolysis of unknown 2 by α-galactosidase produces additional GG.

**Figure 3 biomolecules-13-00148-f003:**
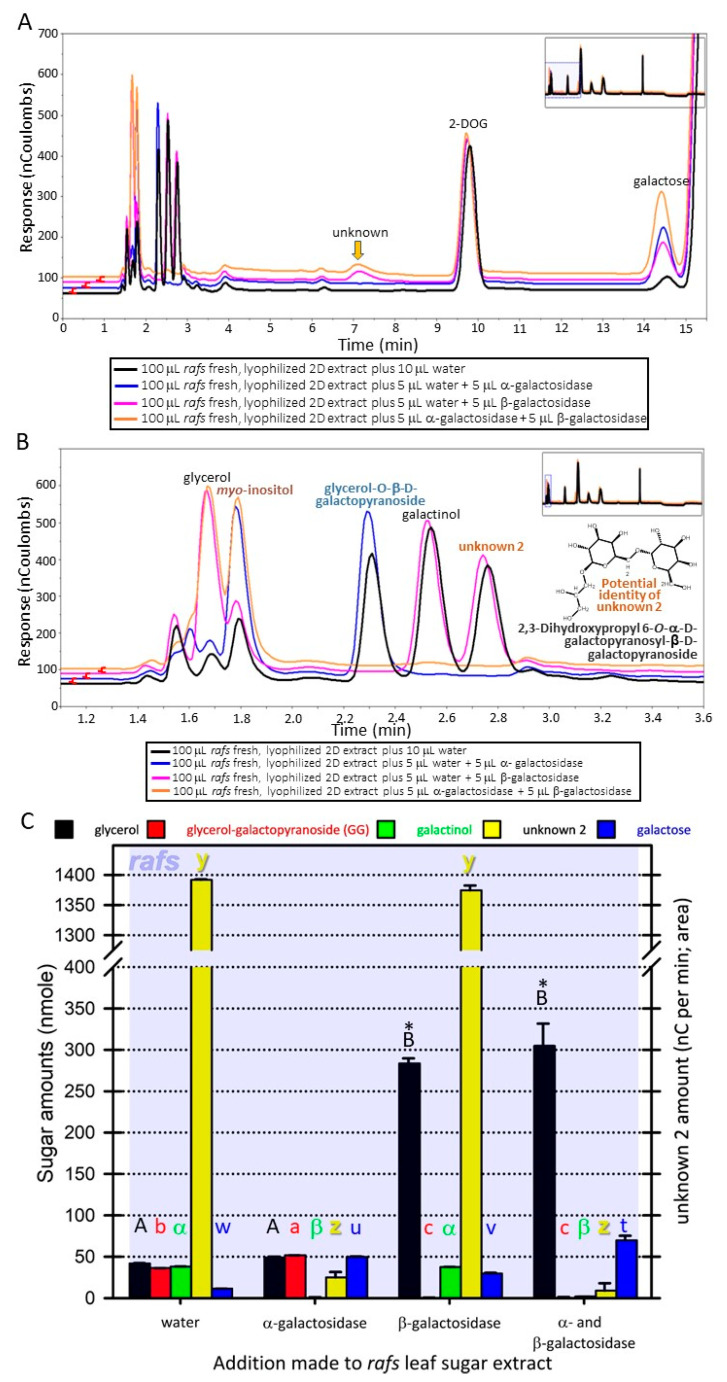
Further verification of the identity of unknowns in maize leaf carbohydrate extracts. The same extracts hydrolyzed in Figure 2 were hydrolyzed again but with both α- and β-galactosidase added simultaneously as another treatment (orange trace). (**A**) Galactose amounts increased above that for unhydrolyzed sample when hydrolyzed by α-galactosidase (blue trace; from galactinol and unknown 2), by β-galactosidase (pink trace; from glycerol-O-β-D-galactopyranoside (GG)) and increased still further upon simultaneous digestion of the sample with both α- and β-galactosidase (orange trace). Simultaneous hydrolysis also eliminated the peaks for GG, galactinol, and unknown 2 (orange trace). (**B**) A closeup of the overlaid traces in A. A potential structural identification of unknown 2 from its mass and patterns of enzyme hydrolysis is presented in the lower right inset. (**C**) An analysis of variance was performed to assess differences in amounts of the metabolites, namely glycerol, GG, galactinol, unknown 2 (in nanocoulombs per min; right axis), and galactose, depending on enzyme treatment. The carbohydrate quantities for all but glycerol were acquired from replicate, 10-fold-diluted, traces (depicted); whereas, to acquire quantities for glycerol that were on the linear portion of the detector response, a 50-fold dilution (not shown) was used. Whenever hydrolysis occurred using β-galactosidase, these glycerol quantities also include glycerol from the commercial preparation. For metabolites, comparisons are made among, not within, enzymatic treatments. Different letters (uppercase, lowercase, or symbol) over the bars among enzymatic treatments denote statistically significant differences among treatments (ANOVA, Scheffe’s multiple comparison test, a = 0.05). The * over glycerol in extracts where the commercial β-galactosidase enzyme was used is predominantly from the enzyme preparation.

**Figure 4 biomolecules-13-00148-f004:**
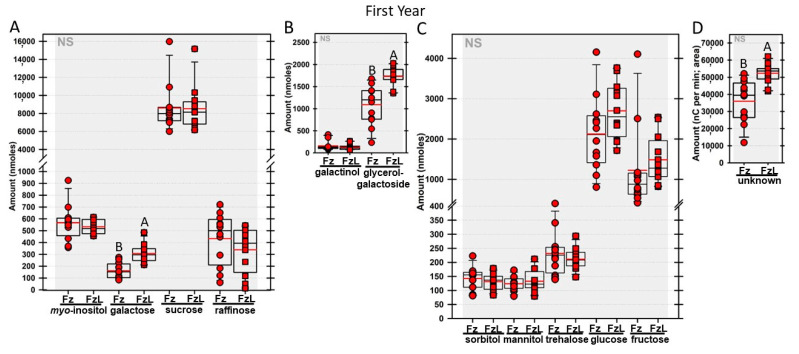
Amounts of soluble carbohydrates from null segregant maize leaf disks sampled in year 1. Carbohydrates were analyzed and quantified from null segregant (NS) maize leaves that were frozen after harvest (Fz; red circles) or were frozen and then lyophilized to dryness (FzL = frozen lyophilized; red squares). For each carbohydrate, the box and whisker plot depicts the mean (red bar) and median (black bar), with the box capturing the interquartile range (25th to the 75th percentile), the whiskers signify the 10th to the 90th percentile, and outliers are dots beyond these whiskers, if any. For carbohydrate amounts (nmoles) from the same-sized disks, different uppercase letters over the box plots between frozen and lyophilized tissues denote statistically significant differences between treatments (ANOVA, Scheffe’s multiple comparison test, α = 0.05). (**A**): *myo*-inositol; galactose; sucrose and raffinose. (**B**): galactinol and glycerol–galactopyranoside. (**C**): Sorbitol; mannitol; trehalose; glucose, and fructose. (**D**): Area under the curve for unknown 2.

**Figure 5 biomolecules-13-00148-f005:**
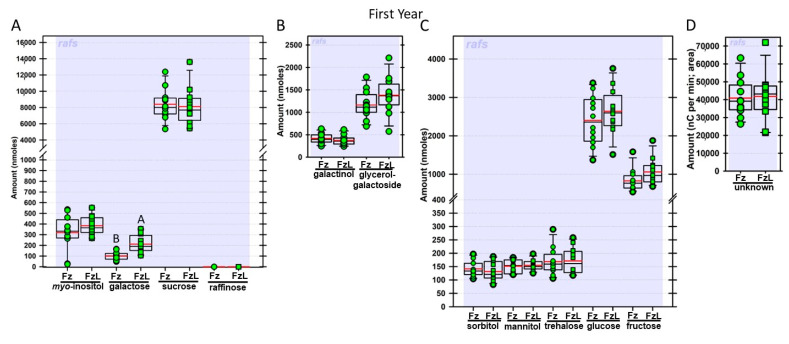
Amounts of soluble carbohydrates from mutant maize leaf disks sampled in year 1. Carbohydrates were analyzed from *raffinose synthase* (*rafs*) maize leaves that were frozen after harvest (Fz; light green circles) or were frozen and then lyophilized to dryness (FzL = frozen lyophilized; light green squares). For each carbohydrate, the box and whisker plot depicts the mean (red bar) and median (black bar), with the box capturing the interquartile range (25th to the 75th percentile), the whiskers signify the 10th to the 90th percentile, and outliers are dots beyond these whiskers, if any. For carbohydrate amounts (nmoles) from the same-sized disks, different uppercase letters over the box plots between frozen and lyophilized tissues denote statistically significant differences between treatments (ANOVA, Scheffe’s multiple comparison test, α = 0.05). (**A**): *myo*-inositol; galactose; sucrose and raffinose. (**B**): galactinol and glycerol–galactopyranoside. (**C**): Sorbitol; mannitol; trehalose; glucose, and fructose. (**D**): Area under the curve for unknown 2.

**Figure 6 biomolecules-13-00148-f006:**
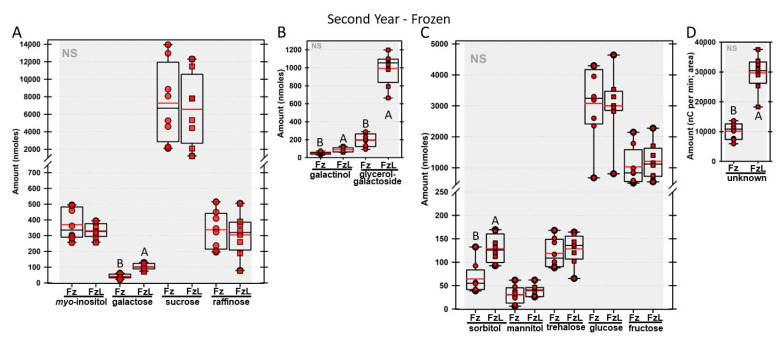
For a second year, carbohydrates were analyzed from null segregant (NS) maize leaves. The disks were frozen after harvest (Fz; red circles) or were frozen and then lyophilized to dryness (FzL = frozen lyophilized; red squares). The same carbohydrates are depicted, and statistically significant differences are determined in the same manner, as in Figure 4.

**Figure 7 biomolecules-13-00148-f007:**
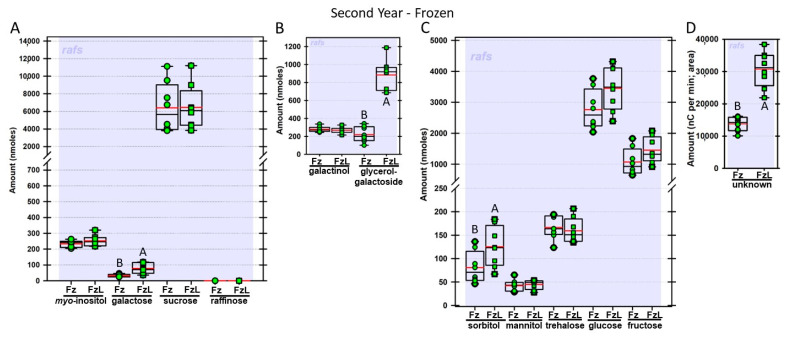
For a second year, carbohydrates were analyzed from *raffinose synthase* (*rafs*) maize leaves. The disks were frozen after harvest (Fz; light green circles) or were frozen and then lyophilized to dryness (FzL = frozen lyophilized; light green squares). The same carbohydrates are depicted, and statistically significant differences are determined in the same manner as in Figure 4.

**Figure 8 biomolecules-13-00148-f008:**
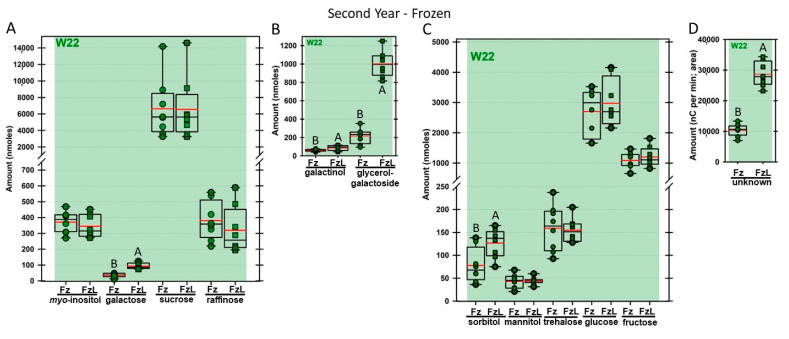
Carbohydrates were analyzed from W22 wild type (W22) maize leaves. The disks were frozen after harvest (Fz; dark green circles) or were frozen and then lyophilized to dryness (FzL = frozen lyophilized; dark green squares). The same carbohydrates are depicted, and statistically significant differences are determined in the same manner as in Figure 4.

**Figure 9 biomolecules-13-00148-f009:**
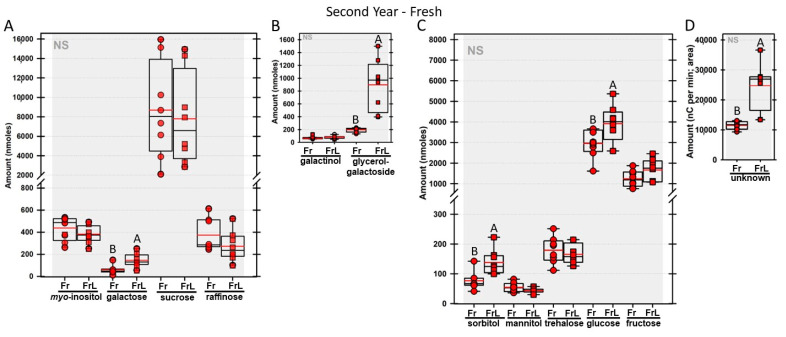
Carbohydrates were analyzed from null segregant (NS) maize leaves. The disks were ground immediately upon harvest (Fr = fresh; red circles) or were lyophilized to dryness without prior exposure to LN_2_ (FrL = fresh lyophilized; red squares). The same carbohydrates are depicted, and statistically significant differences are determined in the same manner, as in Figure 4.

**Figure 10 biomolecules-13-00148-f010:**
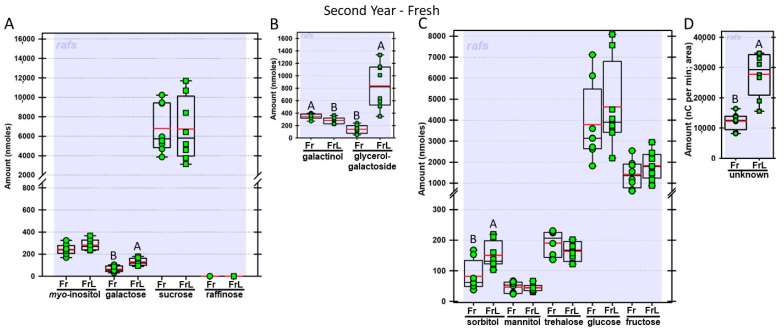
Carbohydrates were analyzed from *raffinose synthase* mutant (*rafs*) maize leaves. The disks were ground immediately upon harvest (Fr = fresh; light green circles) or were lyophilized to dryness without prior exposure to LN_2_ (FrL = fresh lyophilized; light green squares). The same carbohydrates are depicted, and statistically significant differences are determined in the same manner, as in Figure 4.

**Figure 11 biomolecules-13-00148-f011:**
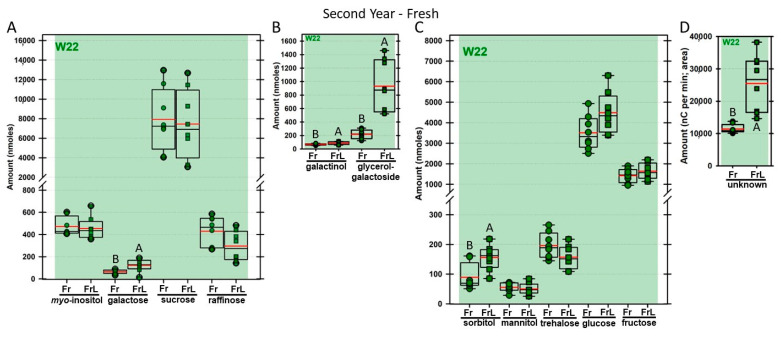
Carbohydrates were analyzed from W22 wild type (W22) maize leaves. The disks were ground immediately upon harvest (Fr = fresh; dark green circles) or were lyophilized to dryness without prior exposure to LN_2_ (FrL = fresh lyophilized; dark green squares). The same carbohydrates are depicted, and statistically significant differences are determined in the same manner, as in Figure 4.

## Data Availability

The leaf extracts from which these data were acquired reside frozen at −20 in the corresponding author’s lab, room 434, Plant Science Building. The chromatographs are present on the computer in the same lab. The assessments leading to the determination of unknown 1 as galactosylglycerol and the mass of unknown 2, are resident in JM’s lab.

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
