# Peer review of "Alterations in Carbohydrate Quantities in Freeze-Dried, Relative to Fresh or Frozen Maize Leaf Disks"

_biomolecules, 2023, doi:10.3390/biom13010148_

Round 1
Reviewer 1 Report
Before storage in refrigerator for further analysis, leaves or roots are usually lyophilized in liquid nitrogen. Thus, it was essential to determine the effect of liquid nitrogen freezing treatment on the concentrations of metabolites. The experiment has great practical significance. The methods were described well.
The introduction was too simple. It should be state out clearly why the authors design and conduct the experiment. What are the scientific questions? And it was lack of the related previous results.
Author Response
Reviewer 1
Open Review
( ) I would not like to sign my review report
(x) I would like to sign my review report
English language and style
( ) English very difficult to understand/incomprehensible
(x) Extensive editing of English language and style required
( ) Moderate English changes required
( ) English language and style are fine/minor spell check required
( ) I don't feel qualified to judge about the English language and style
Both A.B. Downie and L.M. Dirk are native English speakers and have revised the manuscript to clarify points that may have been confusing in the first draft.
|
Yes |
Can be improved |
Must be improved |
Not applicable |
|
Does the introduction provide sufficient background and include all relevant references? |
( ) |
( ) |
(x) |
( ) |
Are all the cited references relevant to the research? |
( ) |
(x) |
( ) |
( ) |
Is the research design appropriate? |
( ) |
(x) |
( ) |
( ) |
Are the methods adequately described? |
(x) |
( ) |
( ) |
( ) |
Are the results clearly presented? |
( ) |
(x) |
( ) |
( ) |
Are the conclusions supported by the results? |
( ) |
( ) |
( ) |
( ) |
Comments and Suggestions for Authors
Before storage in refrigerator for further analysis, leaves or roots are usually lyophilized in liquid nitrogen. Thus, it was essential to determine the effect of liquid nitrogen freezing treatment on the concentrations of metabolites. The experiment has great practical significance. The methods were described well.
The authors are all plant physiologists/molecular biologists interested in the identity and amounts of soluble sugars, particularly the raffinose family oligosaccharides (RFOs), in plant cells. When examining the RFOs in desiccated seeds, (which is one of our research foci) metabolism has already been dampened by the natural loss of water so grinding the seeds in liquid nitrogen is straight forward. However, when we sample leaf cells, it is important that we utilize techniques to minimize alterations to the sugar identities and concentrations in the leaf cells prior to measurement. Reading other’s work, there are some instances where experimenters, for a variety of reasons, plunge the leaves in liquid nitrogen to kill the cells but then lyophilized the leaves prior to storage at -80⁰ C before retrieval at some later point for grinding for sugar extraction. We have, in the past, sampled the leaves, plunged them in liquid nitrogen and then stored them immediately at -80⁰ C. Which storage method is less likely to alter the identity and amount of soluble sugars in the leaves? It was important to assess this. We have samples leaves and ground them in liquid nitrogen after the following methods: 1) Sample the leaves and grind them immediately in liquid nitrogen; 2) Sample the leaves and freeze the in liquid nitrogen before storing them at -80⁰ C; 3) Sample the leaves and put them in the lyophilizer without freezing them first in liquid nitrogen before storing them at -80⁰ C and; 4) Sample the leaves and freeze them in liquid nitrogen, lyophilize them, and then store them at -80⁰ C. Thereafter, those leaves frozen at -80⁰ C were ground in liquid nitrogen and soluble sugars were extracted from the leaves.
The introduction was too simple. It should be state out clearly why the authors design and conduct the experiment. What are the scientific questions? And it was lack of the related previous results.
This is a valid point that has been brought to our attention by all of the reviewers. We have not presented our purpose for these experiments sufficiently well. We have altered the introduction extensively to address this deficiency. The second paragraph of the introduction exemplifies these revisions, “Advancing our understanding of carbohydrate status in maize leaves to create resilience to suboptimal water status requires, first and foremost, analyses that represent, as close as possible, actual leaf cell contents at the time of sampling. Literature reports of maize sugar contents are based on either fresh- (e.g. [13,14] or dry-weight (e.g. [15,16]). The removal of water via lyophilizing after freezing the leaf sample in liquid nitrogen is commonly how a dry weight is obtained prior to the extraction of soluble carbohydrates.” As you will note, we added new references to provide examples of leaf treatment after freezing in liquid nitrogen with or without subsequent lyophilization before storage at -80⁰C prior to sugar extraction.
In the next paragraph we introduced the fact that we have noted changes in certain carbohydrates depending on whether the leaves were lyophilized or not. This is the revised sentence, “In our work extracting carbohydrates from liquid nitrogen frozen or liquid nitrogen frozen and then lyophilized leaf material, we have noted that recovery of certain carbohydrates differed.”
At the end of the introduction we include the following sentence which will help tie our experiments together in the conclusion, “In this comparison, the carbohydrate identity and amounts in leaves frozen in liquid nitrogen was taken as most representative of the true cellular carbohydrate profiles. These were compared to treatments that included lyophilization with or without prior freezing in liquid nitrogen.”
We sincerely hope that these alterations assist in making our reasons for conducting these experiments clear.
Submission Date
15 November 2022
Date of this review
05 Dec 2022 08:40:14
Reviewer 2 Report
1. Please improve the picture definition, such as fig1, fig2.
2. The abstract is suggested to be revised to more accurately highlight the research focus of this experiment or the scientific problems that have been clarified.
3. This study compared the effects of different treatment methods on carbohydrate changes. Under different treatment conditions, other important components such as organic acids, flavonoids, etc. will also have corresponding changes, causing certain positive or negative effects. Why not consider including them in the category of effects on nutritional tissues?
4. Please summarize the research significance of this paper in the summary or conclusion.
5. If possible, it is recommended to add tests to fully prove the further influence of different treatment conditions on the change of carbohydrate content, such as nutritional or functional changes.
6. What is the specific impact of the increase of galactose on the production of leaf tissue?
7. In this study, the galactose content changed significantly before and after different treatments, but the specific source that affected its content was not clear. In fact, in order to ensure the integrity of the study, the source should be clarified. If not, in subsequent studies, the author considered how to clarify its source or what important variables that affected the content (or how to design experiments to prove it?).
Author Response
Reviewer 2
Open Review
( ) I would not like to sign my review report
(x) I would like to sign my review report
English language and style
( ) English very difficult to understand/incomprehensible
( ) Extensive editing of English language and style required
( ) Moderate English changes required
(x) English language and style are fine/minor spell check required
( ) I don't feel qualified to judge about the English language and style
Comments and Suggestions for Authors
- Please improve the picture definition, such as fig1, fig2.
Thank you for pointing out that the clarity of the figure is not helping to project the meaning we are trying to convey. We have reworked all of the chromatographs by expanding the thickness of the chromatographic traces. We have also worked to try to enhance the sharpness of the text and lines in the graphs. I hope that these new presentations are satisfactory.
- The abstract is suggested to be revised to more accurately highlight the research focus of this experiment or the scientific problems that have been clarified.
This is a valid point that has been brought to our attention by all of the reviewers. We have not presented our purpose for these experiments sufficiently well. We have altered the introduction extensively to address this deficiency. The second paragraph of the introduction exemplifies these revisions, “Advancing our understanding of carbohydrate status in maize leaves to create resilience to suboptimal water status requires, first and foremost, analyses that represent, as close as possible, actual leaf cell contents at the time of sampling. Literature reports of maize sugar contents are based on either fresh- (e.g. [13,14] or dry-weight (e.g. [15,16]). The removal of water via lyophilizing after freezing the leaf sample in liquid nitrogen is commonly how a dry weight is obtained prior to the extraction of soluble carbohydrates.” As you will note, we added new references to provide examples of leaf treatment after freezing in liquid nitrogen with or without subsequent lyophilization before storage at -80⁰C prior to sugar extraction.
In the next paragraph we introduced the fact that we have noted changes in certain carbohydrates depending on whether the leaves were lyophilized or not. This is the revised sentence, “In our work extracting carbohydrates from liquid nitrogen frozen or liquid nitrogen frozen and then lyophilized leaf material, we have noted that recovery of certain carbohydrates differed.”
At the end of the introduction we include the following sentence which will help tie our experiments together in the conclusion, “In this comparison, the carbohydrate identity and amounts in leaves frozen in liquid nitrogen was taken as most representative of the true cellular carbohydrate profiles. These were compared to treatments that included lyophilization with or without prior freezing in liquid nitrogen.”
We sincerely hope that these alterations assist in making our reasons for conducting these experiments clear.
- This study compared the effects of different treatment methods on carbohydrate changes. Under different treatment conditions, other important components such as organic acids, flavonoids, etc. will also have corresponding changes, causing certain positive or negative effects. Why not consider including them in the category of effects on nutritional tissues?
We are all plant physiologists/molecular biologists interested in the identity and amounts of soluble sugars, particularly the raffinose family oligosaccharides (RFOs), in plant cells. It is important for us to identify the best means of freezing leaf samples to avoid altering sugar amounts or identities. To do this, we have the very best in carbohydrate separation and identification using HPLC and pulsed electrochemical detection. We are not set up for detection of other metabolites that are outside the focus of our consortium. While we are quite convinced by the observations of the reviewer that alterations in other metabolites would also be influenced by sampling and storage alterations, we are not in a position to assess these with the instrumentation that we possess. Perhaps our identification of soluble sugar alterations due to lyophilization will spur others, investigating other metabolites, to assess their molecules of interest for similar alterations.
- Please summarize the research significance of this paper in the summary or conclusion.
Thank you for suggesting that we could improve on the conclusions. We have tied the conclusion to the revisions we have made in the introduction to more clearly identify our purpose for conducting this research. The end of the introduction now reads, “In this comparison, the carbohydrate identity and amounts in leaves frozen in liquid nitrogen was taken as most representative of the true cellular carbohydrate profiles. These were compared to treatments that included lyophilization with or without prior freezing in liquid nitrogen.” The conclusions now read, “We conclude that maize leaf samples should be collected and frozen in LN2 immediately without subsequent lyophilizing. This is to avoid unwanted changes in soluble carbohydrate quantities, particularly both galactose and glycerol galactopyranoside (GG; potentially also glycerol digalactopyranoside) increases due to lipase action on chloroplast MGDG (potentially also DGDG) membrane components if lyophilized.”
- If possible, it is recommended to add tests to fully prove the further influence of different treatment conditions on the change of carbohydrate content, such as nutritional or functional changes.
While we appreciate the reviewer’s suggestion that plant nutrition would also alter soluble sugar identities and amounts, this is beyond the scope of our current effort which is to identify the best means of storing maize leaves to maintain the soluble sugar identities and amounts prior to extraction.
- What is the specific impact of the increase of galactose on the production of leaf tissue?
We are not entirely sure of what the reviewer is suggesting here. There is no further production of leaf tissue after we have sampled it and removed it from the plant. May we please have a clarification of this issue?
- In this study, the galactose content changed significantly before and after different treatments, but the specific source that affected its content was not clear. In fact, in order to ensure the integrity of the study, the source should be clarified. If not, in subsequent studies, the author considered how to clarify its source or what important variables that affected the content (or how to design experiments to prove it?).
Thank you for your comments above. We are not sure of what alterations of galactose amounts occurred before treatment that the reviewer mentions. Perhaps this is an observation prompted in the first year of our results when rafs plants apparently accumulated more galactose than WT, although this is not apparent in the second year and we did not test comparisons of galactose between the wild type and the rafs mutant. We used the maize rafs mutant to assess whether raffinose hydrolysis was responsible for the increase in galactose that we observed whenever leaves were lyophilized, a post-treatment change. If the increase of galactose was from raffinose hydrolysis, then this should be mitigated in the rafs mutant because there is no raffinose. This was not the case as there is still galactose increasing when rafs leaves are lyophilized. We next focused on the alterations in two unknowns that we saw upon lyophilization of maize leaves. We thought that these might provide a clue as to how the additional galactose was generated upon leaf lyophilization. Using a different HPLC system and tandem mass spectrometry, masses for the two unknowns were obtained which lead to the identification of glycerol-galactose as one unknown and, very probably, glycerol di-galactose as the other. Both are degradation products of chloroplast monogalactosyl diacylglycerol and digalactosyl diacylglycerol membrane components that have had the fatty acid moieties removed by hydrolysis. Without a commercial source of glycerol di-galactose this identification remains tentative. However, we have provided compelling evidence from sequential and simultaneous enzymatic digestion with alpha- and/or beta-galactosidase that both unknowns are entirely consistent with our identifications.
Reviewer 3 Report
This paper entitled “Alterations in carbohydrate quantities in freeze-dried, relative to fresh or frozen maize leaf disks” aimed to solve the problem of pre-extraction of plant leaf samples, anion exchange HPLC and pulsed electrochemical detection. The authors have investigated the discrepancy in carbohydrate amounts between fresh-, frozen-, and lyophilized-leaf disks, and clearly analyzed the effects of different treatment methods before extraction on maize leaf carbohydrates. The purpose of this study is to provide a basis for the selection of preservation methods before leaf sample extraction. The topic was interesting, but some attention should be addressed to improve this paper.
1、Relevant research background needs to be supplemented in INTRODUCTION. And the purpose of the study is not clearly stated.
2、In page 2, line 76. Please explain why you choose the 9th and 10th leaves.
3、In page 8, line 257-258. Please explain "The stability of unknown 2 when treated with β-galactosidase can be explained by the catalytic mechanism of this exo-enzyme."
4、Fig. 2 is blurry and confusing. Please replace more clearer figure.
5、Line 436-439. Conclusion is incomplete and has little correlation with the content of the introduction. The purpose of the research mentioned here is not consistent with the introduction. The paper points out "maize leaf samples should be collected and frozen in LN2 immediately to avoid spurious alterations in soluble carbohydrate quantities." But, the effect of liquid nitrogen freezing on soluble carbohydrates in maize leaves was not explained in the article and there is no direct data to support it. It is suggested to increase the comparison of carbohydrate content between fresh leaves (Fr) and leaves frozen in liquid nitrogen.

Author Response
Reviewer 3
Open Review
( ) I would not like to sign my review report
(x) I would like to sign my review report
English language and style
( ) English very difficult to understand/incomprehensible
( ) Extensive editing of English language and style required
( ) Moderate English changes required
(x) English language and style are fine/minor spell check required
( ) I don't feel qualified to judge about the English language and style
Yes |
Can be improved |
Must be improved |
Not applicable |
|
Does the introduction provide sufficient background and include all relevant references? |
( ) |
( ) |
(x) |
( ) |
Are all the cited references relevant to the research? |
(x) |
( ) |
( ) |
( ) |
Is the research design appropriate? |
( ) |
(x) |
( ) |
( ) |
Are the methods adequately described? |
( ) |
(x) |
( ) |
( ) |
Are the results clearly presented? |
( ) |
(x) |
( ) |
( ) |
Are the conclusions supported by the results? |
( ) |
(x) |
( ) |
( ) |
Comments and Suggestions for Authors
This paper entitled “Alterations in carbohydrate quantities in freeze-dried, relative to fresh or frozen maize leaf disks” aimed to solve the problem of pre-extraction of plant leaf samples, anion exchange HPLC and pulsed electrochemical detection. The authors have investigated the discrepancy in carbohydrate amounts between fresh-, frozen-, and lyophilized-leaf disks, and clearly analyzed the effects of different treatment methods before extraction on maize leaf carbohydrates. The purpose of this study is to provide a basis for the selection of preservation methods before leaf sample extraction. The topic was interesting, but some attention should be addressed to improve this paper.
1、Relevant research background needs to be supplemented in INTRODUCTION. And the purpose of the study is not clearly stated.
This is a valid point that has been brought to our attention by all of the reviewers. We have not presented our purpose for these experiments sufficiently well. We have altered the introduction extensively to address this deficiency. The second paragraph of the introduction exemplifies these revisions, “Advancing our understanding of carbohydrate status in maize leaves to create resilience to suboptimal water status requires, first and foremost, analyses that represent, as close as possible, actual leaf cell contents at the time of sampling. Literature reports of maize sugar contents are based on either fresh- (e.g. [13,14] or dry-weight (e.g. [15,16]). The removal of water via lyophilizing after freezing the leaf sample in liquid nitrogen is commonly how a dry weight is obtained prior to the extraction of soluble carbohydrates.” As you will note, we added new references to provide examples of leaf treatment after freezing in liquid nitrogen with or without subsequent lyophilization before storage at -80⁰C prior to sugar extraction.
In the next paragraph we introduced the fact that we have noted changes in certain carbohydrates depending on whether the leaves were lyophilized or not. This is the revised sentence, “In our work extracting carbohydrates from liquid nitrogen frozen or liquid nitrogen frozen and then lyophilized leaf material, we have noted that recovery of certain carbohydrates differed.”
At the end of the introduction we include the following sentence which will help tie our experiments together in the conclusion, “In this comparison, the carbohydrate identity and amounts in leaves frozen in liquid nitrogen was taken as most representative of the true cellular carbohydrate profiles. These were compared to treatments that included lyophilization with or without prior freezing in liquid nitrogen.”
We sincerely hope that these alterations assist in making our reasons for conducting these experiments clear.
2、In page 2, line 76. Please explain why you choose the 9th and 10th leaves.
Thank you for this idea. We have used the ninth and tenth leaves because they are still vigorous at the time of tasseling and by doing so, we avoid senescing leaves. Additionally, the ninth and tenth leaves, at this time of the plant’s development, are not so small as to prevent taking two circular samples besides each other on either side of the main vein. We avoid the main vein because we are interested in the soluble carbohydrate concentration of the cells of the photosynthetic cells of the leave and so, despite the maize leaf parallel venation, wish to minimize the assimilate coming from these sources through the main vein. Here is how we worded this with the new text in red, “Harvest, freezing, and freeze-drying: Upon tasseling, pairs of 3.7 cm (dia.; 11.62 cm2) disks were cut from the blades of each leaf of a plant on either side of the main vein 20 cm down from the tip (duplicates; Fig. 1). These disks were taken from the 9th and 10th leaf from the same plant (replications) and one disk from the 9th and its replicate disk from the 10th leaf underwent the same treatment. The 9th and 10th leaves are vigorous at tasseling and large enough to allow acquisition of two circular samples beside each other on either side of the main vein. The main vein is avoided to minimize the assimilate coming through it. The first year the treatments constituted frozen in liquid nitrogen…”
3、In page 8, line 257-258. Please explain "The stability of unknown 2 when treated with β-galactosidase can be explained by the catalytic mechanism of this exo-enzyme."
Thank you for pointing out this point for clarification. Our tentative identification of unknown 2 as glycerol di-galactose and its probably origin from hydrolysis of free fatty acids from digalactosyl diacylglycerol would then dictate that the galactose moieties would be linked to glycerol as: glycerol (beta) galactose (alpha) galactose. Trying to hydrolyse this molecule with beta galactosidase which is an exo-enzyme capable of hydrolysing beta linked galactoses ONLY if they are at the end of an oligomer will fail because the linkage exposed to the enzyme is an alpha linkage, while the beta linkage is shielded from the enzyme by the intervening, alpha linked galactose. We have added text to this section to clarify this statement (see the red text below).
“The stability of unknown 2 when treated with beta-galactosidase can be explained by the catalytic mechanism of this exo-enzyme [21] i.e. the beta-linked galactose is shielded from the enzyme by the alpha-linked galactose.”
4、Fig. 2 is blurry and confusing. Please replace more clearer figure.
Thank you for pointing out that the clarity of the figure is not helping to project the meaning we are trying to convey. We have reworked all of the chromatographs by expanding the thickness of the chromatographic traces. We have also worked to try to enhance the sharpness of the text and lines in the graphs. I hope that these new presentations are satisfactory.
5、Line 436-439. Conclusion is incomplete and has little correlation with the content of the introduction. The purpose of the research mentioned here is not consistent with the introduction. The paper points out "maize leaf samples should be collected and frozen in LN2 immediately to avoid spurious alterations in soluble carbohydrate quantities." But, the effect of liquid nitrogen freezing on soluble carbohydrates in maize leaves was not explained in the article and there is no direct data to support it. It is suggested to increase the comparison of carbohydrate content between fresh leaves (Fr) and leaves frozen in liquid nitrogen.
We understand the concerns expressed by the reviewer. However, the freshly harvested leaves were pulverized by exposing them to liquid nitrogen. There is no means by which we can ensure leaf cell rupture without freezing the leaves in liquid nitrogen. Our objective with these experiments was to determine if lyophilization, with or without prior freezing in liquid nitrogen introduced differences in soluble sugars. In this we succeeded to demonstrate that galactose consistently increases in the lyophilized leaves of three genotypes of maize and we have provided strong evidence suggesting that the source of the galactose comes from chloroplast galactosyl diacylglycerols.
Round 2
Reviewer 3 Report
The article has been improved greatly by the modification, and I think the article in present form can be accepted.